# HiPDERL: An Improved Implementation of the PDERL Viewshed Algorithm and Accuracy Analysis

Haozhe Cheng and Wanfeng Dou *

School of Computer Science and Technology, Nanjing Normal University, Nanjing 210023, China
* Correspondence: douwanfeng@njnu.edu.cn

**Abstract:** Terrain viewshed analysis based on the digital elevation model (DEM) is of significant application value. A lot of viewshed analysis algorithms have been proposed, including R3 as the accurate one and others as efficient ones. The R3 algorithm is accurate because of its comprehensive but time-consuming computation, while the others are efficient due to proper approximation. However, no algorithm is capable of taking advantage of both until one algorithm is proposed, which is based on a 'proximity-direction-elevation' (PDE) coordinate system and named the PDE spatial reference line (PDERL) algorithm. The original research proves the PDERL algorithm is perfectly accurate by theory and experimental results, in comparison with R3 as standard, and even more efficient than R3. However, the original research does not mention the cases where the observer is placed on grid points, and the original implementation does not produce very accurate results in practice. It is important to find out and correct the errors. In this paper, a checking algorithm for PDERL is proposed to allow further investigation of errors. With the fundamental ideas of PDERL unchallenged, an improved implementation of the PDERL algorithm is proposed, named HiPDERL. By experimental results, this paper proves HiPDERL utilizes the potential of PDERL on accuracy at the cost of a little efficiency when the observer is placed on grid points.

**Keywords:** terrain viewshed analysis; viewshed computation; PDERL algorithm; digital elevation model

## 1. Introduction

Terrain visibility analysis is an important part of digital terrain analysis. The viewshed in a terrain refers to a distinct region of the terrain that can be seen at one or more specific positions within a certain range on the terrain, and viewshed analysis is the process to compute such viewshed, with the elevation value of a given location based on the digital elevation model (DEM) data [1]. Terrain viewshed analysis based on the digital elevation model is widely used in archaeological research [2], path planning [3], siting optimization [4], landscape analysis [5], military analysis [6], security monitoring [7], or combination with other grids for terrain description [8,9].

Terrain viewshed analysis computes the viewshed of a viewpoint at a certain position on the terrain. This position is taken as the viewpoint with an observer on it, and all positions within a certain range around the viewpoint are taken as the target points. By doing so, the viewshed analysis of the viewpoint is thus transformed into the intervisibility computation between the viewpoint and each target point. The general method for computing intervisibility is to connect the viewpoint and the target point to create a line-of-sight (LOS) and determine whether the target point is visible to the viewpoint, by studying whether the line-of-sight is blocked by the terrain [10], which is the fundamental basis of most viewshed computation algorithms. Commonly, considering the height of the observer, the actual elevation of the viewpoint may be slightly higher than the elevation of the position itself on the DEM.

A lot of algorithms have been proposed for viewshed analysis, including R3 as the accurate one and others as efficient ones. The R3 algorithm [6] is known for its accuracy.

According to the R3 algorithm, the intersection point of the line-of-sight and the grid line indicates the terrain traversed by the line-of-sight and the elevation value of which is obtained by the interpolation of the two adjacent grid points. The accuracy of the R3 algorithm is secured by independent interpolation on each intersection point of each line-of-sight, resulting in time-consuming computation. In order to solve the efficiency problem of the R3 algorithm, a lot of algorithms have been proposed, including the R2 algorithm, the XDraw algorithm [6] and the reference plane algorithm [11]. By multiplexing and approximating, the time complexity of computation is reduced at the cost of some accuracy. Many improved methods are proposed based on these algorithms as well. To take the XDraw algorithm as an example, it takes only two points near the target point as the reference points to compute the viewshed of the viewpoint. Compared with the R3 algorithm, the XDraw algorithm greatly simplifies the computation process and improves the computational efficiency at the cost of certain accuracy. In order to improve the accuracy of the XDraw algorithm, Izraelevitz et al. traced back along the line-of-sight and included more points in computation [12]. Zhi et al. introduced the historical minimum visual elevation of each target point into computation as additional data [13]. Zhu et al. proposed the HiXDraw algorithm, using contribution points instead of reference points in XDraw as the basis for computing the visibility of the target points, thus avoiding the interference of irrelevant visible points (referred to as 'chunk distortion' in their work) on computational results [10]. To further improve the efficiency of the algorithms, many parallel computing methods have been proposed and applied to viewshed analysis as well, showing great potential [14]. Apart from these traditional algorithms or their enhanced versions, Tabik et al. [15] employ horizon calculation methods [16] in viewshed analysis. By dividing terrain into angle-based sectors, the viewshed result is turned into visible areas rather than visible points. This method opens a door for the following work [17,18] that focuses on efficiency.

This paper deeply studies another algorithm proposed by Wu et al. [19], based on a 'proximity-direction-elevation' (PDE) coordinate system and named the PDE spatial reference line (PDERL) algorithm. The PDERL algorithm stores some computational results in a data structure named the reference line, in order to present the exact elevation values of different positions with the greatest capability to block the LOS in all directions by accurate interpolation computation rather than an approximation to near grid point. Compared with the R3 algorithm, the efficiency of the PDERL algorithm is improved by data reuse. Compared with other algorithms, the accuracy of the PDERL algorithm is not compromised. In a word, the PDERL algorithm is able to take advantage of both and solve the problem of the inevitable trade-off between accuracy and efficiency. The original research proves the PDERL algorithm is perfectly accurate by theory and experimental results, in comparison with R3 as standard, and even more efficient than R3. However, the original research does not mention the cases where the observer is placed on grid points, and the original implementation does not produce very accurate results in practice. It is important to find out and correct the errors. In this paper, a checking algorithm for PDERL is proposed to find out problems that impair accuracy. The problems prove to be some implementation flaws and the float number accuracy problem. With the fundamental ideas of PDERL unchallenged, an improved implementation of the PDERL algorithm is proposed to solve or alleviate these problems, named as HiPDERL. In Section 2, the related work is introduced, namely the viewshed algorithms involved in this paper. In Section 3, the improvement work is introduced, including the process and problems of the PDERL algorithm, as well as the checking algorithm for PDERL and the improved PDERL implementation proposed in this paper. In Section 4, experiments are carried out to show the problems above through specific examples, then the accuracy and efficiency of the improved PDERL implementation are compared with other viewshed algorithms on various terrains. A summary is given in Section 5.

## 2. Related Works

### 2.1. Viewshed Algorithms

Existing viewshed algorithms mainly include the accurate algorithm R3 and other more efficient algorithms. The R3 algorithm ensures high accuracy through complex interpolation computation without approximation methods, while other algorithms achieve better efficiency through multiplexing with approximation to near grid point around the LOS, at the cost of certain accuracy.

The R3 algorithm is an accurate algorithm based on-line-of sight. The R3 algorithm takes the viewpoint as the start point and each target point as the end point to create a straight line as the line-of-sight between them. Each line-of-sight intersects several grid lines on the DEM and produces intersection points. For convenience, these points are termed LOS-intersection ones, because there are other kinds of intersection points in this paper. The R3 algorithm obtains the elevation value of the LOS-intersection points through linear interpolation of the elevation value of two adjacent grid points and then determines whether the LOS-intersection points block the line-of-sight from the target point. For each target point, the R3 algorithm performs the processes mentioned above on LOS-intersection points on the line-of-sight, until the line-of-sight is determined to be blocked or not.

Other efficient algorithms record the computational results of each target point and reuse them through approximate methods to process other target points. Traditional algorithms include the R2 algorithm, the XDraw algorithm and the reference plane algorithm. To take the XDraw algorithm as an example, the XDraw algorithm determines the visibility of the target point only by the LOS-intersection point closest to the target point. The elevation value of the LOS-intersection point depends on the two adjacent grid points, which are the only points that the XDraw algorithm needs to process. With procedures simplified, the efficiency of the XDraw algorithm is therefore greatly improved in computational efficiency compared with the R3 algorithm. The XDraw algorithm records the reference height of the target point, which of the visible target point is recorded as the elevation value of the target point itself, and which of the invisible target point is recorded as the lowest elevation value that the target point needs to reach to be visible.

HiXDraw, an improved algorithm of XDraw, believes that the reference height of invisible target points only transmits the elevation value of other visible target points, and possibly involves some visible target points irrelevant to the LOS in the computation. Such a phenomenon is referred to as 'chunk distortion'. The HiXDraw algorithm introduces the concept of the contribution point. Each invisible target point has one or two visible contribution points that contribute to its visibility computation, and each visible target point has one contribution point itself. In the actual process, the HiXDraw selects two reference points according to the XDraw procedure and obtains two to four contribution points of both reference points, two of which are selected to apply the reference plane algorithm. Compared with the XDraw algorithm, the HiXDraw algorithm eliminates chunk distortion through the steps mentioned above and reaches higher accuracy at the cost of some efficiency.

To measure the time complexity of these algorithms, let $n$ be the radius of the area for target points, and the number of target points that a viewshed algorithm needs to process is at the scale of $n^2$ on the area. The time complexity of the R3 algorithm for processing one target point depends on the distance between the viewpoint and the target point, reaching the scale of $O(n)$, so the overall time complexity of the R3 algorithm is $O(n^3)$. The other efficient algorithms simplify the computation at the cost of some accuracy. The time complexity of processing one target point is usually $O(1)$, so the overall time complexity is $O(n^2)$. The PDERL algorithm introduced below is able to achieve the overall time complexity of $O(n^2)$ as well, without compromising the computational accuracy.

### 2.2. PDERL Algorithm

#### 2.2.1. Fundamental Ideas

The PDERL algorithm is similar to the R3 algorithm in that they both use interpolation to determine whether the LOS is blocked by the LOS-intersection points, so the result is accurately figured out as well. Rather than using approximation methods, the PDERL algorithm reorganizes the computation procedures within R3 and realizes data reuse. The actual work to be performed for R3 and PDERL is the same. With these fundamental ideas, the PDERL algorithm is able to greatly improve efficiency without loss of accuracy.

According to the original ideas of the PDERL algorithm, the viewpoint can be placed everywhere on the DEM. In order to compare the PDERL algorithm with other viewshed algorithms that place the viewpoint on grid points, in this paper the viewpoints of the PDERL algorithm are placed on grid points as well, which allows the computational results to be checked by the checking algorithm for PDERL proposed in this paper. Please view Section 3.1 for details.

For the LOS-intersection points on the LOS in the R3 algorithm, the PDERL algorithm divides them into two types: LOS-intersection points generated by the LOS and horizontal or vertical grid line, for convenience referred to as horizontal-LOS intersections and vertical-LOS intersections, respectively, in this paper. The PDERL algorithm then divides the DEM into four overlapping regions. The target points on the right, left, upper and lower sides of the viewpoint are defined to be in Region I, Region II, Region III and Region IV, respectively, as shown in Figure 1. Each target point belongs to one or two regions if the viewpoint is on a grid point.

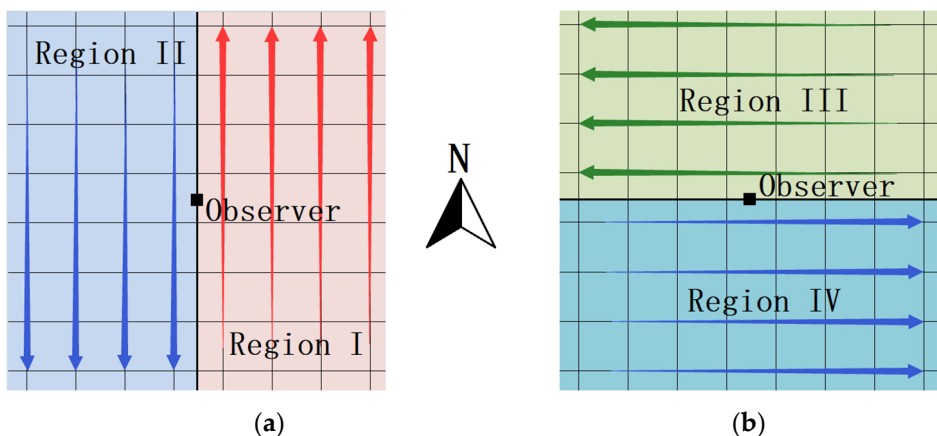

       (**a**)                                            (**b**)

**Figure 1.** All target points are divided into four overlapping regions. Points are processed line by line, and lines closer to the ovserver are processed before the ones farther to the observer: (**a**) Region I and Region II; (**b**) Region III and Region IV.

The PDERL algorithm takes two steps to compute the viewshed of a viewpoint. For the first step, the algorithm processes Region I and Region II, and unilaterally computes the visibility of each target point only by determining whether the LOS is blocked by vertical-LOS intersections. Target points are considered visible if they prove to be visible in the first step. For the second step, the algorithm processes Region III and Region IV and does the same work with horizontal-LOS intersections only. Target points are considered invisible if they prove to be invisible in the second step, so the possible wrong visibility conclusions obtained in the first step are corrected in the second step. The visibility computation through the two steps above is generally comprehensive because the PDERL algorithm takes all LOS-intersection points into account. However, when the observer is on a grid point, for the target points in the same column of the viewpoint, the algorithm simply skips these target points. Please view Section 3.2.1 for details.

The algorithm constructs the corresponding PDE coordinate system for the visibility computation of each region. The algorithm takes the viewpoint as the origin, sets the

*X*-axis on the DEM plane according to the direction corresponding to the region (right, left, up and down), then sets the *Y*-axis by rotating the *X*-axis anti-clockwise to 90°, and finally sets the *Z*-axis in the direction where elevation value increases. Then, the XYZ coordinate system is converted into the PDE coordinate system according to the following method, as shown in Figure 2. The letters PDE, respectively, represent the proximity-direction-elevation coordinate system of the target point relative to the viewpoint. Their transformation relationships with XYZ coordinate system are shown in Equations (1)–(3), adapted from Ref. [19]. The algorithm computes the viewshed based on the PDE coordinate system.

$$p = 1/x \tag{1}$$

$$d = \tan(\alpha) = yp = y/x \tag{2}$$

$$e = \tan(\gamma) = zp = z/x \tag{3}$$

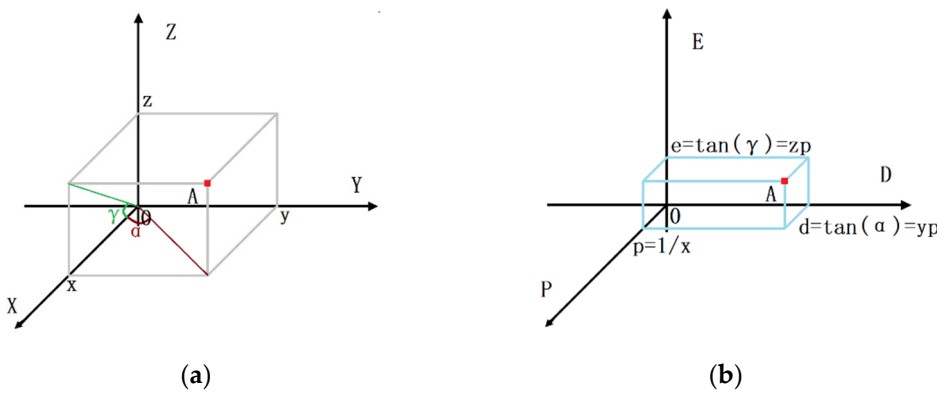

**Figure 2.** Point A in different coordinate systems: (**a**) The XYZ coordinate system; (**b**) The PDE coordinate system.

The PDE coordinates are vital parts of the PDERL algorithm. The letter *p* stands for proximity. In the same region, the algorithm always processes target points with the same and larger *p* value in batches, that is, one row or one column of target points closer to the viewpoint. The letter D stands for direction. When processing a row or a column of target points with the same *p* value, the algorithm always follows the order of increasing the *d* value. The letter E stands for elevation and is of most significance in this algorithm. Though the *e* value serves the same purpose as the elevation measurement used in the R3 algorithm, according to Equation (2), they are actually not the same, as the *e* value in the calculation is irrelevant to the distance between the viewpoint and the target point. The *e* value quantifies the ability of the LOS-intersection points in the same direction D to block the LOS, in another word, the ability of target points not to be blocked. By quantitative computation and numerical comparison, the *e* value is used to compute the visibility of one target point and is stored for subsequent computation.

The PDERL algorithm computes the visibility of all target points in one region by constructing and updating reference polylines, as shown in Figure 3. To take Region I with the corresponding direction of right as an example, the algorithm firstly considers the target points on the $C_1$ column as visible, because there are no vertical-LOS intersections that may block the LOS between the viewpoint and the target points on the $C_1$ column. Then, a reference polyline reflecting the elevation of each target point on column $C_1$ is produced in a plane rectangular coordinate system with a horizontal axis D and a vertical axis E. The reference polyline reflects the maximum *e* value among all LOS-intersection points in any specified direction D. For the next step, the algorithm puts the next column (i.e., $C_2$ column) into the D-E coordinate system and computes the visibility of each target point on the $C_2$ column with the help of the reference polyline. In any given direction D, if the point on the $C_2$ column is higher than the point on the reference polyline (i.e., the corresponding

*e* value is larger), it means the corresponding terrain on column $C_2$ is not blocked and therefore visible. Otherwise, it means the terrain corresponding to column $C_2$ is blocked and invisible. After completing the computation of each target point corresponding to the $C_2$ column polyline, the algorithm combines the two polylines by the, respectively, higher parts as a new reference polyline to show the occlusion caused by the $C_1$ column and the $C_2$ column together in each direction D (as shown in Figure 4). The steps mentioned above are repeated to compute the visibility of each target point on the $C_3$ column and so on until the visibility computation of all target points in the first region is completed.

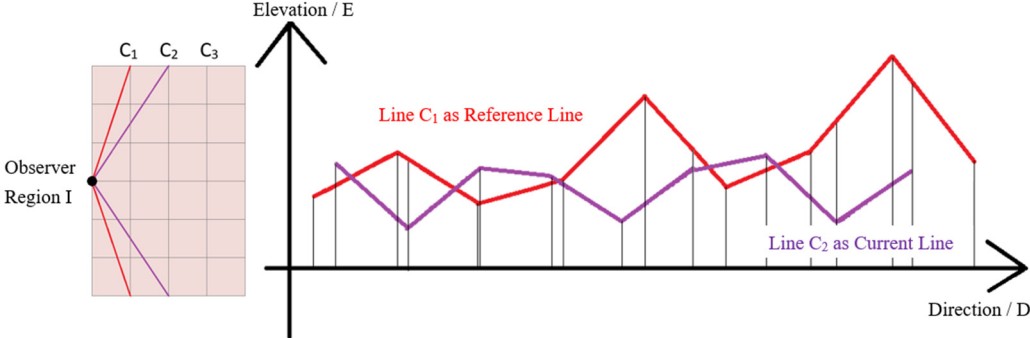

**Figure 3.** In the D-E coordinate system, while processing target points corresponding to line $C_2$, line $C_1$ is the reference line, and line $C_2$ is the current line.

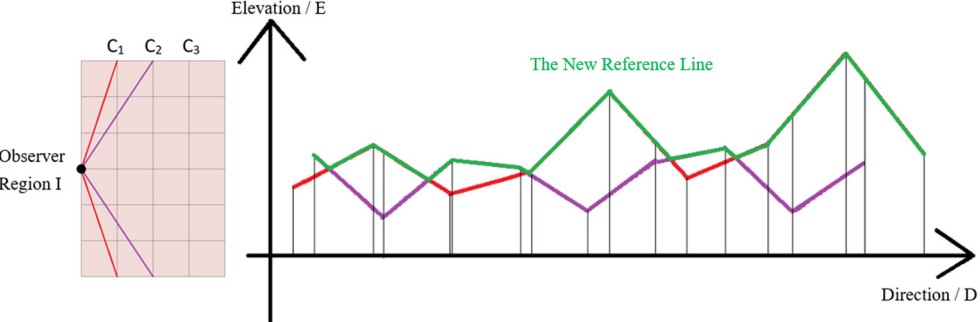

**Figure 4.** In the D-E coordinate system, the higher parts of $C_1$ and $C_2$ are combined as the new reference line for further computation.

The PDERL algorithm completes the visibility computation of each target point in each region according to the steps above. Each target point is considered visible only if it is considered visible in all the regions it belongs to.

Although the horizontal coordinate D corresponds to uneven actual distance, the original literature on the PDERL algorithm has proved that in the D-E coordinate system, the *e* value between every two points on each polyline still changes linearly as the horizontal coordinate D changes [19].

### 2.2.2. Implementation Details

The actual computation is demonstrated in Figure 5, where $L_R$ is the reference polyline and $L_C$ is the currently processing row or column polyline. For convenience, the $L_C$ polyline is referred to as the current polyline. The D and E coordinates of each inflection point on the current polyline $L_C$ (i.e., the target points) are directly calculated from DEM data, while that of the reference polyline $L_R$ are stored in a linked list. In the implementation of PDERL, only the head node of the linked list stores the D and E coordinate of the first point on the reference polyline $L_R$, while the other nodes store the D coordinate of the inflection point, and the slope value *a* of the segment between the point itself and the previous point. Let the slope value *a* stored for one point be $a_j$ and assume the D and E coordinate of the point be $d_j$ and $e_j$, coordinates of the previous point be $d_{j-1}$ and $e_{j-1}$, then the value of $a_j$ is shown

in Equation (4), which may be distinguished from the slope value in Figure 5. The vertical coordinate E of each inflection point on the reference polyline stored in this way needs to be calculated by traversing the linked list.

$$a_j = \frac{e_j - e_{j-1}}{d_j - d_{j-1}} \tag{4}$$

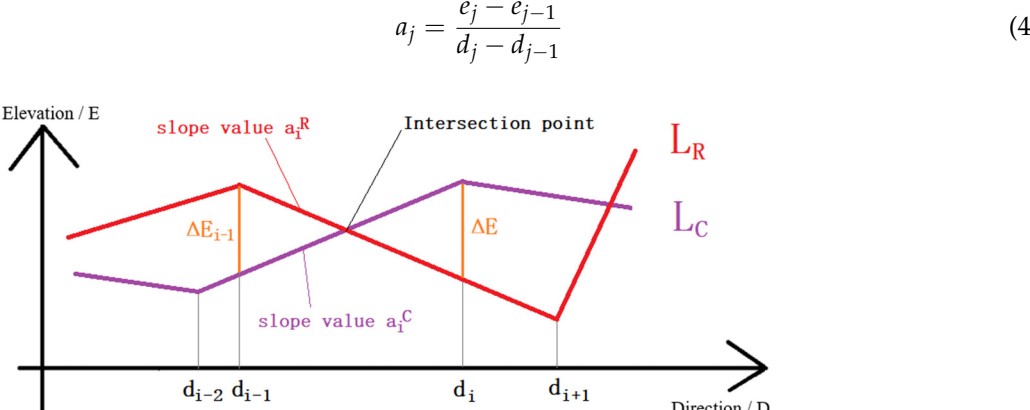

**Figure 5.** In the D-E coordinate system, by creating subsections and comparing the $e$ value on the two polylines at the end of each subsection, the visibility of the corresponding terrain on line $L_C$ is determined.

In the D-E coordinate system, for any given direction D, if there are corresponding points on $L_R$ and $L_C$, whose $e$ values are assumed to be $e_R$ and $e_C$, respectively, the difference value $\Delta E$ is calculated as shown in Equation (5), adapted from Ref. [19]. Let the visibility of the terrain corresponding to the point on current polyline $L_C$ be $r_C$, the value of $r_C$ is shown in Equation (6), where zero means invisible and one means visible in the corresponding region. Please notice that in this paper, the criteria are set that the $\Delta E$ value must be above zero for the corresponding terrain to be visible. When the terrain on $L_C$ happens to be the target point, the visibility of the target point is computed.

$$\Delta E = e_C - e_R \tag{5}$$

$$r_C = \begin{cases} 0, & \Delta E_C \leq 0 \\ 1, & \Delta E_C > 0 \end{cases} \tag{6}$$

The actual calculation is based on subsections. As shown in Figure 5, for $L_R$ and $L_C$ in the D-E coordinate system, the algorithm divides the two polylines into many subsections according to the horizontal coordinate D of all inflection points on the two polylines. This ensures that in each subsection, there are only one or zero intersection points, which means that when moving positively along the D axis, there are only one or zero changes in the visibility of the terrain corresponding to the current polyline within a subsection. For each subsection, the algorithm does two parts of work. The first part is to calculate the $\Delta E$ value at the end of the subsection, in order to determine the visibility of the corresponding terrain on $L_C$ at the end of the subsection. The second part is to compare the $\Delta E$ value at the end of the subsection with that of the last subsection, in order to determine whether the two polylines have an intersection within this subsection and figure out the coordinates of the intersection if it exists, for preparation to update the reference polyline.

In the implementation of the PDERL algorithm, let the horizontal coordinate D at the end of the subsection be $d_i$ and the $\Delta E$ value be $\Delta E_i$, the horizontal coordinate D at the beginning of the subsection be $d_{i-1}$ and the $\Delta E$ value be $\Delta E_{i-1}$, the slopes of $L_R$ and $L_C$ within the subsection be $a_i^R$ and $a_i^C$, respectively, (obtained by referring to the reference polyline and the DEM, respectively), then the value of $\Delta E_i$ is shown in Equation (7), adapted from Ref. [19]. The $\Delta E_i$ value calculated in this way depends on the value of $\Delta E_{i-1}$. If there is an intersection within a subsection, i.e., $\Delta E_i$ and $\Delta E_{i-1}$ have opposite positive and negative values, the coordinates of the intersection are then calculated. Let the horizontal

coordinate D of the intersection be d', and the value of d' is shown in Equation (8), adapted from Ref. [19]. For the intersection point to be stored in the updated reference polyline, apart from the horizontal coordinate D mentioned above, the slope value *a* is required as well, which is directly obtained from the not-updated reference polyline.

$$\Delta E_i = \Delta E_{i-1} + \left( a_i{}^C - a_i{}^R \right)(d_i - d_{i-1}) \tag{7}$$

$$d' = d_i - \frac{\Delta E_i}{a_i{}^C - a_i{}^R} \tag{8}$$

In addition, the implementation of the PDERL algorithm realized an error-avoiding mechanism to deal with $\Delta E$ with too small an absolute value. In a subsection, If the absolute value of $\Delta E$ at the beginning of the subsection is less than a certain small value and there exists an intersection, the algorithm will consider the horizontal coordinate D of the intersection as equal to the coordinates at the beginning of the subsection in order to avoid errors. After the reference line is updated, the intersection point will share the same horizontal coordinate D with the starting point of the subsection, resulting in the redundancy of the reference line. Please view Section 3.2.2 for relevant content.

### 3. HiPDERL of Improvements on PDERL Algorithm

The original research proves the PDERL algorithm is perfectly accurate by theory and experimental results, in comparison with R3 as standard, and even more efficient than R3. However, the original research does not mention the cases where the observer is placed on grid points, and the original implementation does not produce very accurate results in practice. It is important to find out and correct the errors. Since the PDERL algorithm, as well as R3, investigates all intersection points made by the LOS and grid lines, a checking algorithm for PDERL is proposed to find out all the problems, substantially caused by the differences between R3 and PDERL in practical detail. All problems found are shown in this section. To deal with all these problems, a new implementation algorithm named HiPDERL is proposed in order to achieve better accuracy.

In another word, the checking algorithm is proposed to check the original PDERL and find what is different from R3 in practical detail, and HiPDERL as another implementation is proposed to minimize the differences, thus achieving better accuracy.

### 3.1. Checking Algorithm for PDERL

According to Equation (5), the $\Delta$E value is obtained by subtracting the *e* values of the corresponding points on $L_R$ and $L_C$, i.e., $e_R$ and $e_C$, where $e_R$ is the maximum *e* value of all LOS-intersection points in the direction D, and $e_C$ is the *e* value of the target point when the corresponding point on $L_C$ is the target point. Since the horizontal coordinate D of the target point is exactly the direction of the LOS, the R3 algorithm could be transformed into a checking algorithm for PDERL based on the LOS. In the R3 algorithm, assuming the horizontal and vertical coordinate differences between the target point and the viewpoint as m and n, respectively, there are m − 1 horizontal-LOS intersections and n − 1 vertical-LOS intersections on the LOS. Let the elevation values of the target point and the viewpoint (with the height of the observer) on the DEM be $h_T$ and $h_S$, respectively, and assume each horizontal/vertical-LOS intersection as the *i*th point closest to the viewpoint, with its elevation value on DEM as $h_i$. Let the theoretical $\Delta E$ value of the target point be $\Delta E_A$ when the PDERL algorithm is processing Region I and Region II, or $\Delta E_B$ when it is processing Region III and Region IV, then the value of $\Delta E_A$ and $\Delta E_B$ is shown in Equations (9) and (10).

$$\Delta E_A = e_C - e_R = \frac{h_T - h_S}{m} - \max\left( \frac{h_i - h_s}{i} \right) \ , \ i = 1, 2, \ldots, m - 1 \tag{9}$$

$$\Delta E_B = e_C - e_R = \frac{h_T - h_S}{n} - \max\left( \frac{h_i - h_s}{i} \right) \ , \ i = 1, 2, \ldots, n - 1 \tag{10}$$

In theory, the value of $\Delta E_A$ or $\Delta E_B$ is supposed to be equivalent to those $\Delta E$ values figured out by the PDERL algorithm, so the checking algorithm is able to verify the computational result of the PDERL algorithm. Figure 6 shows how the checking algorithm checks the value of the value $\Delta E_B$ above. The checking algorithm calculates the maximum $e$ value of all the horizontal-LOS intersections, i.e., the value of $e_R$ of the coordinating point on the reference polyline $L_R$ in the D-E coordinate system. The checking algorithm also calculates the $e$ value of the target point, i.e., the value of $e_L$ of the coordinating point on the current polyline $L_C$ in the D-E coordinate system, while the target point is considered as in Region IV rather than in Region I. Therefore, the value of $\Delta E_B$ can be obtained by the difference between the value of $e_R$ and $e_C$ above, and then the corresponding $\Delta E$ value in the PDERL algorithm is checked.

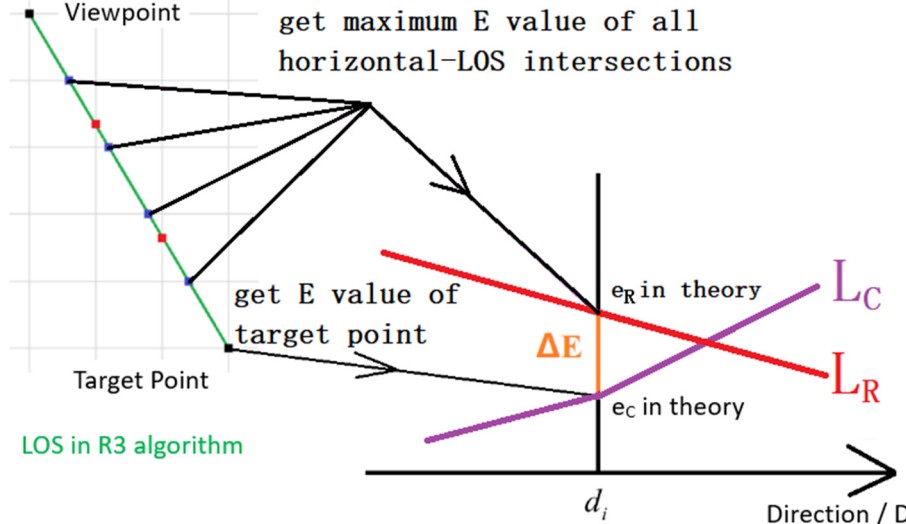

**Figure 6.** The principle of the checking algorithm for PDERL based on the line-of-sight, where horizontal-LOS intersections refer to the intersections of the LOS and the horizontal grid line.

The differences in the computational results between the PDERL algorithm and the R3 algorithm are studied by checking the value of $\Delta E$. With the support of the checking algorithm, the problems within the original PDERL implementation are found and shown in this paper. The checking algorithm itself is not for viewshed computation.

### 3.2. Existing Problems

3.2.1. Problem 1: Unprocessed Column

The first problem lies in an unexpected flaw. According to the design of the implementation of PDERL, the viewpoint can be placed off the grid points. However, when the viewpoint is placed on a grid point, the PDERL is not able to compute the visibility of the target points in the column where the viewpoint is located. The reasons are as follows. When the PDERL algorithm processes Region I and Region II, the visibility of the target points mentioned above is not computed, because they are not on the left or right side of the viewpoint. When the PDERL algorithm processes Region III and Region IV, the algorithm will only consider some of the visible target points as invisible. Therefore, there is no chance to compute the visibility of the target points that shares the same column with the viewpoint. The original implementation initializes the visibility of all target points as invisible before the PDERL algorithm begins, so the target points mentioned above are considered invisible once and for all. To solve this problem, independent visibility computation of these target points is added to the normal workflow of PDERL in this paper. The independent visibility computation is similar to R3, while its complexity is O(1) because the line composed of the target points allows data reuse. The solution to this

problem is completely irrelevant to other problems since these target points are completely out of the PDERL algorithm.

### 3.2.2. Problem 2: Flaws within the Error Avoidance Mechanism

The second problem lies in that there are mistakes within the error avoidance mechanism in the PDERL implementation. This is another flaw as well. As shown in Figure 7, when $L_R$ and $L_C$ have an intersection within a subsection, and if the absolute value of $\Delta E$ is too small, the error avoidance mechanism within the original implementation considers the horizontal coordinate D of the intersection as equivalent to $d_A$, as shown in Figure 7. However, in some cases, the mechanism mistakenly considers the horizontal coordinate D of the intersection as the same as $d_B$, resulting in an error that may be transmitted via the fundamental data reuse of the PDERL algorithm. In order to solve this problem, this mechanism is remade in this paper.

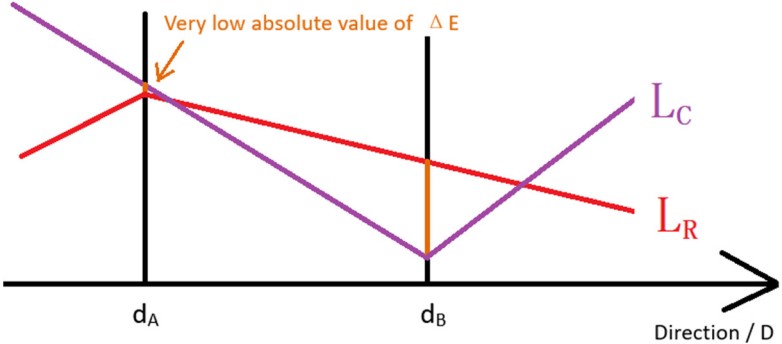

**Figure 7.** The case to trigger the original error avoiding mechanism within PDERL.

### 3.2.3. Problem 3: Float Number Accuracy

With the checking algorithm, the implementation of PDERL proves to have some float number errors generated by machine operation, and these errors are transmitted through data reuse. As an example of the errors, $(2.5 \times 0.4 - 10 \times 0.1)$ may not be calculated as absolute zero but as a small positive value. The original error-avoiding mechanism proved not effective against these errors.

Apart from the fundamental data reusing method within the PDERL algorithm that simplifies the visibility computation of subsequent target points by referring to the reference polyline, the PDERL algorithm applied data reusing methods to data structure and the calculation of $\Delta E$ value. For the data reusing method for data structure, the PDERL uses the linked list to store the reference polyline, whose nodes store the horizontal coordinate D value of the corresponding inflection point, and the slope value $a$ between the inflection point itself and the previous one. Therefore, the $e$ value of each inflection point needs to be calculated on a lot of float numbers by traversing the linked list. For the data reusing method for the calculation of $\Delta E$ value, it is shown in Equation (7) that the calculation of each $\Delta E$ value depends on the previously calculated $\Delta E$ value.

Although these methods manage to simplify the computation, reduce the number of variables used and achieve better efficiency, when an error occurs in calculating the $e$ value or the $\Delta E$ value, for example, when some absolute zero values are calculated to be a non-zero number with very small absolute value, the data reuse may transmit the error in the two ways as follows:

(1) The vertical coordinate E of the inflection point on the reference line needs to be obtained by traversing the linked list. It is possible that an error caused by float number operation occurs while traversing the linked list in order to calculate the vertical coordinate E, and this error may become the basis for calculating subsequent $e$ values. In this way, the error is passed on.

(2) The calculation of each ΔE value depends on the previous ΔE value. Therefore, any error that occurred within the calculation of a ΔE value results in the transmission of this error among all the following ΔE values.

The original error-avoiding mechanism may respond to the errors mentioned above, but is not designed to eliminate them, and therefore fails to prevent some errors from interfering with the computational result. This paper proposes an improved implementation of the PDERL algorithm, named HiPDERL to better solve these problems. Please view Section 4.2.2 for examples of the float number accuracy problems.

### 3.3. HiPDERL: An Improved PDERL Implementation Algorithm

HiPDERL, an improved implementation of the PDERL algorithm is proposed to solve the problems. HiPDERL simply applies the corrections mentioned in Section 3.2.1 to solve problem 1. It would be like processing the target points within the PDERL system and processing the unprocessed column independently afterwards. In addition, the improved implementation realizes a new error-avoiding mechanism to deal with the error within the calculation of ΔE, thus solving problem 2. If the absolute value of ΔE is smaller than a fixed small value, the ΔE value will be considered as an error of machine operation and set to exactly zero. The improved implementation will also check and eliminate the redundant inflection points on the reference line after the calculation on each current line $L_C$ is completed, so as to reduce the probability for errors to occur.

For problem 3, i.e., the float number accuracy problem, HiPDERL makes changes to the data reusing methods on data structure and the calculation of ΔE value, in order to minimize the generation or transmission of errors. The details within HiPDERL are introduced in this section. The fundamental ideas and procedures of the PDERL algorithm on data reuse remain unchanged. The terrain with the highest $e$ value in each direction of the viewpoint is still stored by the reference polyline to simplify the visibility computation of each following target point.

The improved implementation changes the data structure of the reference polyline, using the nodes of the linked list to store the horizontal coordinate D and the vertical coordinate E of the corresponding point so that the $e$ value of the corresponding point could be obtained without traversing the linked list, thus avoiding the possible error caused by float number operation in the traversal process. Due to these changes, if the intersection of the two polylines exists in any subsection, the vertical coordinate E of the intersection must be calculated as well, as one more value to be calculated in addition to the horizontal coordinate D of the intersection and the ΔE value at the end of the subsection. The improved implementation changes the way ΔE value is calculated as well, so the ΔE value no longer depends on the previous ΔE value, thus avoiding the transmission of relevant errors.

The details of the calculation are as follows. The improved implementation uses more data from the current polyline rather than the reference line for calculation, in order to minimize dependency on previous results that may cause accuracy problems.

The horizontal coordinate D of the intersection of the two polylines is still calculated according to Equation (8). However, since the linked list no longer stores the slope value $a$, this value needs to be calculated on relative inflection points of the reference polyline.

The vertical coordinate E of the intersection of the two polylines is calculated as follows: let the horizontal coordinate D of the intersection obtained according to Equation (8) be d′, and the vertical coordinate E be $e′$. Assume the horizontal coordinate D of the inflection point on the current polyline (i.e., the target point) closest to the intersection point and in the negative direction of axis D as $d_{NC}$, and the vertical coordinate E as $e_{NC}$. Assuming the slope value of the current polyline at the intersection is $a_i{}^C$, then the value of E′ is shown in Equation (11). The value of $a_i{}^C$ is calculated with the DEM data.

$$e′ = e_{NC} + a_i{}^C * (d′ - d_{NC})  \qquad (11)$$

The ΔE value at the end of each segment is still obtained by the difference between $e_R$ and $e_C$ according to Equation (5). However, since the previous ΔE value is no longer de-

pended on, the calculation of the ΔE value needs to be discussed under different conditions that the inflection point at the subsection is on the reference line or the current line.

When the inflection point of the subsection's end is on the reference polyline, the value of $e_R$ can be directly obtained from the reference polyline, and the value of $e_C$ needs to be calculated. Let the horizontal coordinate D at the end of the subsection be $d_i$, and the ΔE value there be $\Delta E_{iR}$. Assume the horizontal coordinate D of the inflection point on the current polyline (i.e., the target point) closest to the end of the subsection and in the negative direction of axis D as $d_{NC}$, and the vertical coordinate E as $e_{NC}$. Assuming the slope value of the current polyline at the end of the subsection be $a_i{}^C$, then the value of $\Delta E_{iR}$ is shown in Equation (12). The value of $a_i{}^C$ is calculated with the DEM data.

$$\Delta E_{iR} = e_C - e_R = \left( e_{NC} + a_i{}^C * (d_i - d_{NC}) \right) - e_R \tag{12}$$

When the inflection point of the subsection's end is on the current polyline, the value of $e_C$ can be directly obtained from the current polyline, and the value of $e_R$ needs to be calculated. Let the horizontal coordinate D at the end of the subsection be $d_i$, and the ΔE value there be $\Delta E_{iC}$. Assume the horizontal coordinate D of the inflection point on the reference polyline closest to the end of the subsection and in the negative direction of axis D as $d_{NR}$, and the vertical coordinate E as $e_{NR}$. Assuming the slope value of the reference polyline at the end of the subsection be $a_i{}^R$, then the value of $\Delta E_{iC}$ is shown in Equation (13). The value of $a_i{}^R$ is calculated with the reference polyline data.

$$\Delta E_{iC} = e_C - e_R = e_C - \left( e_{NR} + a_i{}^R * (d_i - d_{NR}) \right) \tag{13}$$

These are the new ways in which the values are calculated. In a word, to solve the float number accuracy problem caused by the traversal of the linked list, the improved implementation limits the generation and transmission of float number error by using the new data structure to represent the reference line and remaking a new error-avoiding mechanism. To solve the same problem caused by the calculation of the ΔE value, the improved implementation avoids using the previously calculated ΔE value in the calculation of the ΔE value. As mentioned in Section 3.2.3, the original implementation achieves better efficiency with some methods. The HiPDERL applies some other methods to ensure accuracy rather than the original efficient ones. Therefore, compared with the PDERL algorithm, the HiPDERL algorithm is expected to trade limited efficiency cost for better accuracy performance.

## 4. Experimental Results

### 4.1. Experimental Environments

Computational accuracy and efficiency experiments on different DEMs were conducted in this section.

There are two groups of DEMs used for the experiments. The first group is used for demonstration of problems within the PDERL, which involves one single 200 × 200 DEM obtained by sampling the original 2000 × 2000 DEM of Malaga, Spain at the horizontal and vertical interval of 10 points. The original DEM is adapted from the input DEM data provided by the implementation from Ref. [16], where the upper part is mountainous and the lower right part is a large zero-elevation area. The resolution of the original 2000 × 2000DEM is 10 m × 10 m, so the resolution of the 200 × 200 DEM used in this paper is 100 m × 100 m. Please notice that with such size and resolution, this DEM could hardly be meaningful for any practical application. It is used for a better demonstration of the problems only. The second group is used for accuracy and efficiency comparison experiments among various viewshed algorithms, which involves 3601 × 3601 DEMs of different terrains including mountains (N28E097), hills (N41E119) and plains (N34E114) from the ASTER GDEM. The resolution of these DEMs is 30 m × 30 m.

The accuracy experiment in this section (Section 4.2) took the R3 algorithm as the baseline algorithm. Problems 1, 2 (Section 4.2.1) and problem 3 (Section 4.2.2) of the PDERL algorithm are demonstrated on the first group of DEM, i.e., the 200 × 200 DEM of Malaga, Spain. The results by R3, PDERL, and HiPDERL are shown for comparison. The overall accuracy was compared among PDERL, HiPDERL, R3, XDraw, and HiXDraw on the second group of DEMs of different terrain (Section 4.2.3). XDraw and HiXDraw are used as references, in order to show the results of a traditional multiplexing algorithm and its enhanced version. The efficiency experiment in this section (Section 4.3) compared the time consumed by various algorithms in the accuracy comparison experiment in Section 4.2.3, respectively.

The hardware environment of this experiment is a computer with Windows 10 operating system, Intel (R) core (TM) i7-9750 h CPU at 2.60 GHz frequency, and 16GB of physical memory. These experiments were carried out in a single-threaded C++ environment, and the corresponding garbage collection mechanism was implemented for the linked list used by the PDERL algorithm. Considering that the HiPDERLand PDERL implementations realized the error-avoiding mechanism to eliminate float number errors, this experiment also configured similar mechanisms for R3, XDraw, and HiXDraw.

## 4.2. Experiments on Accuracy

### 4.2.1. Analysis of Implementation Flaws

The original error-avoiding mechanism has some implementation flaws, and cannot compute the visibility of the target points that share the same column with the viewpoint. An example that reflects the two problems at the same time is shown as follows. Figure 8 shows the case that R3 and PDERL were used to compute the viewshed of the viewpoint located at row 96, column 176 on the 200 × 200 DEM of Malaga, Spain mentioned above. Around the viewpoint are hills with escalating elevation from south to north, but the viewpoint is exactly surrounded by relatively higher grid points, so the viewshed on this point is not very large. In Figure 8, a red pixel represents a visible target point, and a blue pixel represents an invisible one. Figure 8a,b is the result of the R3 and HiPDERL algorithm, and Figure 8c,d is the result of the PDERL algorithm. The position of the viewpoint is marked with a white cross. Please notice that the problems could be on all DEMs, while the problems on this DEM may be better demonstrated.

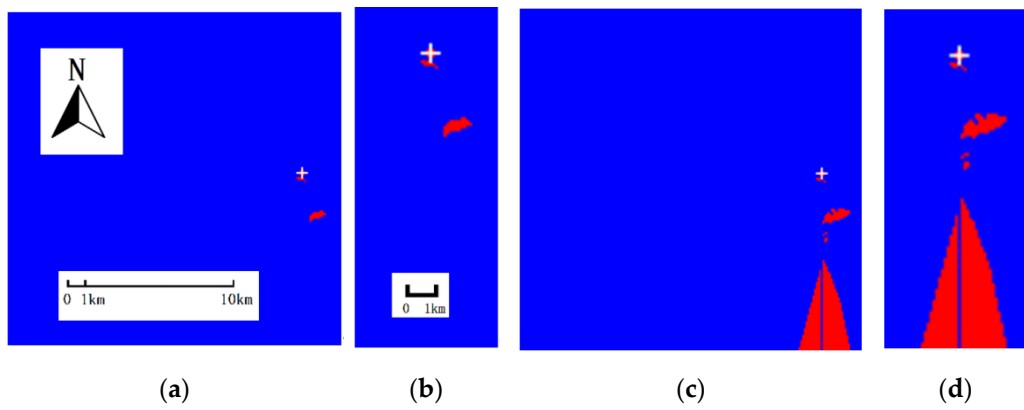

| (a) | (b) | (c) | (d) |

**Figure 8.** The viewshed results of the viewpoint at row 96, column 176 on the DEM of Malaga, Spain by different algorithms. Observer is marked with a white cross. Red area is visible and blue area is invisible. (**a**) R3, while a totally same result by every cell is figured out using HiPDERL; (**b**) Part of (**a**), row 80–200, column 150–200; (**c**) HiPDERL; (**d**) Part of (**c**), row 80–200, column 150–200.

From the figure, it can be found that the PDERL algorithm cannot compute the viewshed of the target points in the column where the viewpoint is located, and they are initialized as invisible at the beginning. In addition, the result of the PDERL algorithm has more visible parts than the result of the R3 algorithm, which is due to the dislocation of the

reference polyline caused by the original error avoiding mechanism and error accumulation caused by data reuse. The HiPDERL algorithm figured out the same viewshed result as the R3 algorithm after correcting the error-avoiding mechanism and adding independent computation of the target points in the column where the viewpoint is located. This case partly shows the consequence of the problems within the original implementation.

### 4.2.2. Analysis of Float Number Accuracy

The PDERL implementation suffers from the float number accuracy problem caused by machine operation errors. Figure 9 shows the case that R3 and PDERL were used to compute the viewshed of the viewpoint located at row 183, column 185 on the $200 \times 200$ DEM of Malaga, Spain. Around the viewpoint is a water area with an elevation of zero. In Figure 9, the red or blue pixel represents visible or invisible corresponding target points. Figure 9a,b is the result of the R3 and HiPDERL algorithm, and Figure 9c,d is the result of the PDERL algorithm. The position of the viewpoint is marked with a white cross. Please notice that the problems could be on all DEMs, while the problems on this DEM may be better demonstrated.

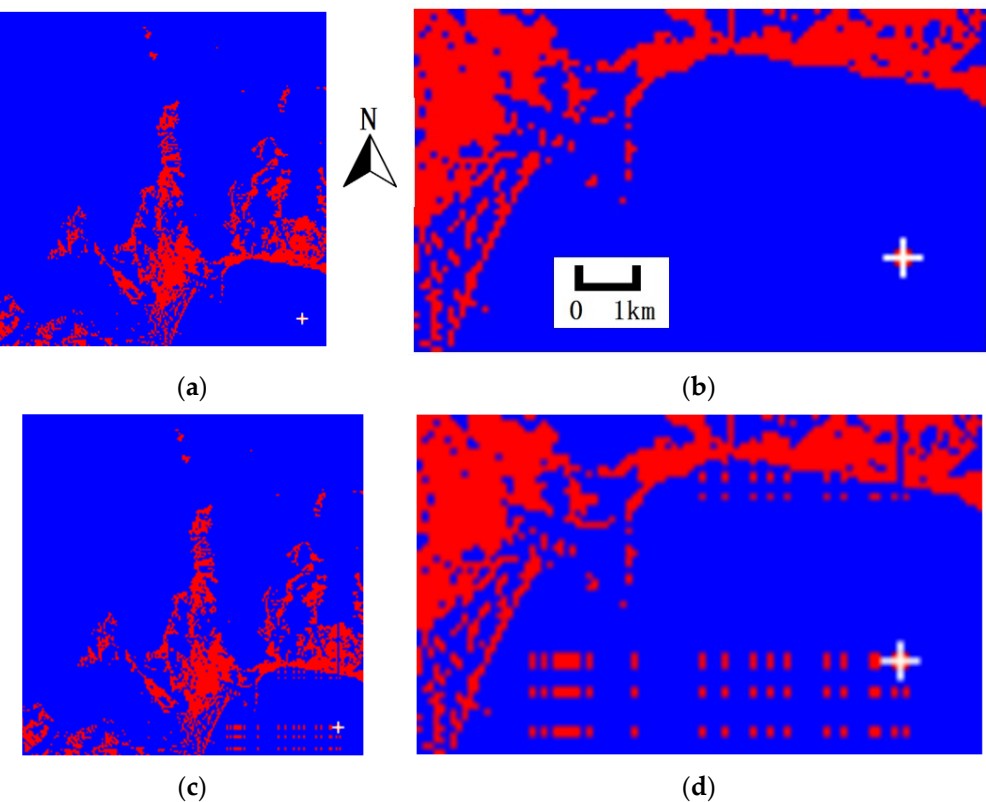

**Figure 9.** The viewshed results of the viewpoint at row 183, column 185 on the DEM of Malaga, Spain by different algorithms. Observer is marked with a white cross. Red area is visible and blue area is invisible. (**a**) R3, while a totally same result by every cell is figured out using HiPDERL; (**b**) Part of (**a**), row 140–200, column 100–200; (**c**) HiPDERL; (**d**) Part of (**c**), row 140–200, column 100–200.

In this example, the height of the observer at the viewpoint was set to zero. Figure 9 shows that the result of the PDERL algorithm is inconsistent with the result of the R3 algorithm in the lower right corner, which is similar to noise. It is mentioned above that the lower right part of the DEM is a zero-elevation area. The viewpoint, the target point, the noise points and the whole LOS are all located in this zero-elevation area. Since the height of the observer was zero as well, the R3 algorithm figured out that the elevation of the viewpoint and all the LOS-intersection points is zero, so the target point was determined to be invisible, as shown in Figure 10a. When the PDERL algorithm

traversed the reference polyline, due to the error generated by float number calculation, the $e$ value supposed to be zero was calculated as a negative number with a very small absolute value. Because the $e$ value of the target point on the current polyline was zero, the $\Delta E$ value was positive and the target point was determined to be visible. At this time, the slope value $a$ stored in the subsequent nodes on the reference polyline was zero, so the $e$ value of the subsequent points of the reference polyline was calculated as this error again, as shown in Figure 10b. Although this noise problem can be neglected by changing the criteria of visibility determination in Equation (6), this is not recommended because in that case, some visible points would be considered invisible and be noise points again due to the same reason.

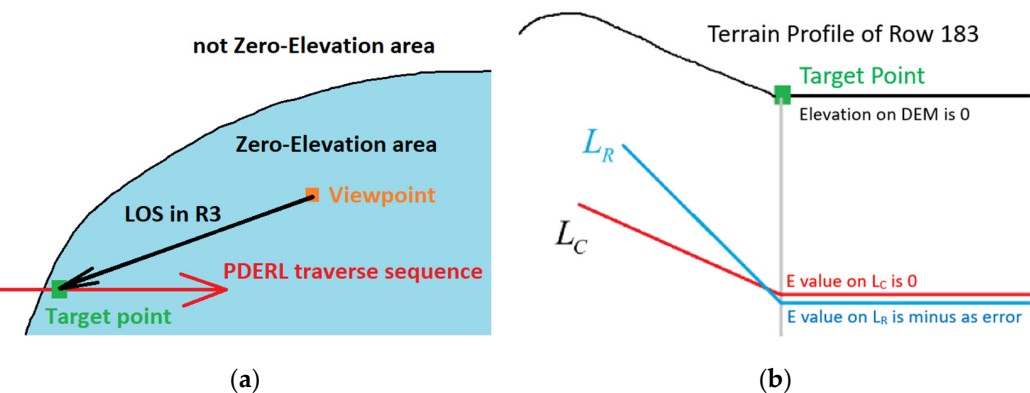

(**a**)                                 (**b**)

**Figure 10.** Reasons for the problem shown in Figure 9. (**a**) The DEM demonstration around the viewpoint at row 183, column 185; (**b**) The terrain profile of row 183. To calculate the visibility of the target point (marked with green in both figures), the PDERL algorithm must traverse almost the whole of row 183.

In general, the float number accuracy problem of the PDERL algorithm came from two aspects: the inherent error of the floating number operation and the error transmission caused by the implementation of the PDERL algorithm. The HiPDERL algorithm proposed in this paper is intended to solve the problems above. Compared with the PDERL, the result of the HiPDERL greatly reduced the amount of noise.

4.2.3. Comparison Experiment on Accuracy

This experiment takes 100 points as the horizontal and vertical interval on the $3601 \times 3601$ DEM of mountains, hills, and plains, thus obtaining 1225 points as viewpoints and computed their viewshed within a radius of 100 points for each viewpoint with various algorithms. Since the radius was taken as 100 points, the side length of each viewshed square is 201 points, which means for each viewpoint, the visibility of a total of 40, 400 target points within the viewshed square was computed. The number of all target points involved in this experiment is therefore very large, which is believed to eliminate possible contingencies. To use PDERL with problems unsolved as a comparison, the original version and the version free of problem 1 are selected. Problems 2 and 3 may seem separate, but they both involve small float absolute values and may interfere with each other, so they are regarded as one here. With the result of the R3 algorithm as baseline, this experiment compared the result of PDERL (original version and the version free of problem 1), HiPDERL, XDraw, and HiXDraw. Since the R3 algorithm was used as the baseline algorithm, the viewshed result by each algorithm inconsistent with the R3 algorithm of each viewpoint was regarded as errors. The experiment counted the total number of errors in the viewshed results of each viewpoint with different algorithms, and the results are shown in Table 1.

**Table 1.** Total error number and error rate on different DEMs with different algorithms (Total cell number: 49,490,000).

| Algorithms\DEMs | Mountains | Hills | Plains |
|---|---|---|---|
| PDERL (Original) | 42,545 (0.860%) | 108,061 (2.183%) | 115,522 (2.334%) |
| PDERL (Problem 1 solved) | 18,055 (0.365%) | 112,756 (2.278%) | 123,052 (2.486%) |
| HiPDERL | 73 (0.001%) | 429 (0.009%) | 313 (0.006%) |
| XDraw | 114,949 (2.323%) | 129,131 (2.609%) | 86,397 (1.746%) |
| HiXDraw | 110,491 (2.233%) | 102,810 (2.077%) | 42,725 (0.863%) |

The experimental result shows that the original PDERL implementation is not more accurate than XDraw due to the problems mentioned above. The solution to problem 1 is helpful but not critical. For differences on different terrains, PDERL performed more accurately in the mountain area. In hill and plain areas, PDERL is not more accurate than the traditional XDraw algorithm. The HiPDERL algorithm shows the accuracy that a PDERL implementation should have. Experiments in the original research of PDERL claim that the PDERL algorithm is perfectly accurate. This may be the result of not placing the observer on the grid point. When the observer is on the grid point, many grid points will be in the same line, which is not critical for R3 because results on different points are computed separately, and few random float number errors will not be harmful. Things become very different for PDERL. Due to the fundamental idea of data reuse, errors (if any) will be certainly passed on and impair the accuracy of further computation. The case that the observer is on the grid point will make more coincidences that a target point is exactly invisible, which may be computed as visible by machine operation. So, it is important to adjust the error-avoiding mechanism and add new modifications to the original PDERL implementation.

Further detailed results of HiXDraw and PDERL (with problem 1 solved) are shown in Tables 2 and 3. Compared with an enhanced and more accurate version of XDraw, the errors caused by the problems within the original PDERL may be better illustrated. For the cases where one algorithm makes over 100,000 errors on certain terrain, the original PDERL has more viewpoints than HiXDraw with an error rate below 0.1%. This indicates that the original PDERL is able to be accurate for more viewpoints. However, for the viewpoints with an error rate above 1%, those of the original PDERL (77 in hill area and 75 in plain area) are much more than that of HiXDraw (21 in mountain area and 19 in hill area).

**Table 2.** Total error number and viewpoint number with different error rate with PDERL (with problem 1 solved) (Total viewpoint number: 1225).

| DEMs | Mountains | Hills | Plains |
|---|---|---|---|
| Total error number | 18,055 | 112,756 | 123,052 |
| Viewpoint number(Perfectly accurate) | 511 | 147 | 136 |
| Viewpoint number(Error rate below 0.1%) | 615 | 610 | 511 |
| Viewpoint number(Error rate 0.1%–0.5%) | 78 | 302 | 400 |
| Viewpoint number(Error rate 0.5%–1%) | 15 | 89 | 103 |
| Viewpoint number(Error rate above 1%) | 6 | 77 | 75 |

**Table 3.** Total error number and viewpoint number with different error rate with PDERL (with problem 1 solved) (Total viewpoint number: 1225).

| DEMs | Mountains | Hills | Plains |
|---|---|---|---|
| Total error number | 110,491 | 102,810 | 42,725 |
| Viewpoint number(Perfectly accurate) | 4 | 25 | 84 |
| Viewpoint number(Error rate below 0.1%) | 401 | 482 | 910 |
| Viewpoint number(Error rate 0.1%–0.5%) | 706 | 626 | 291 |
| Viewpoint number(Error rate 0.5%–1%) | 93 | 98 | 22 |
| Viewpoint number(Error rate above 1%) | 21 | 19 | 2 |

Further investigation is carried out into the viewpoints with an error rate above 1%. There are cases where error points gather or scatter for both algorithms, including the cases with the most errors. The PDERL errors spread among lines from the first error point. The reasons are explained above. On the other hand, as an enhanced version of XDraw, HiXDraw suffers from accumulated errors positioned behind an error point, giving a radical distribution in the zenithal view image. Considering the statistics shown in the tables, HiPDERL provides a solution to the accuracy problems.

*4.3. Comparison Experiment on Efficiency*

In this experiment, the time consumed by each algorithm in the accuracy comparison experiment above was counted, i.e., the time consumed by each algorithm including the R3 algorithm to compute the viewshed of 1225 viewpoints within the range of 100 points on each DEM. The results are shown in Table 4.

**Table 4.** Time consumed on different DEMs with different algorithms (seconds).

| Algorithms\DEMs | Mountains | Hills | Plains |
| --- | --- | --- | --- |
| R3 | 126.028 | 92.207 | 62.311 |
| PDERL | 40.178 | 39.978 | 39.594 |
| HiPDERL | 45.710 | 45.197 | 44.725 |
| XDraw | 37.071 | 36.903 | 37.091 |
| HiXDraw | 45.275 | 44.292 | 44.108 |

The experimental result shows that XDraw achieved the highest computational efficiency, while the efficiency of PDERL and HiPDERL proved not significantly inferior to that of the XDraw family algorithms and was higher than that of the R3 algorithm. The efficiency of the HiPDERL was slightly lower than that of the PDERL. This is because the HiPDERL changes some data reusing methods in the PDERL are prone to generate or transmit errors, and also makes some computations more complex, thus increasing the computation time.

The experimental result also shows that the time consumed by the R3 algorithm in different terrain was not constant, while the time consumed by other algorithms was relatively constant. One possible explanation is that when the R3 algorithm is to determine whether the LOS is blocked and once the LOS is blocked, the algorithm may determine the target point as invisible and immediately interrupt the computation, so the actual time complexity may be lower than the theoretical value. It is usually difficult to predict whether and where the LOS is blocked because this varies among different maps. The steps of other algorithms are relatively simple, so the actual workload on different terrain was relatively fixed as well.

## 5. Conclusions

This paper pointed out that when the observer is not on grid points, the original PDERL algorithm does not produce very accurate results in practice. After the analysis of the PDERL algorithm, a checking algorithm for the PDERL algorithm was proposed in order to find out the problems. With all problems found, a new implementation algorithm named HiPDERL, based on the fundamental ideas of PDERL, was proposed to solve the problems. Experiments showed that HiPDERL achieved better accuracy than the original PDERL implementation at the cost of a little efficiency. The future work is to apply the HiPDERL algorithm to the computation of total viewshed. There exist very efficient total viewshed algorithms, but no baseline algorithm is used for comparison on accuracy because the traditional accurate algorithms R3 is less efficient for total viewshed analysis and the cost in time will be immeasurable. HiPDERL greatly reduces the cost in time and maintains the computational accuracy in comparison with R3. Therefore, HiPDERL could be a potential baseline algorithm to assess the accuracy of existing and upcoming total viewshed algorithms.

**Author Contributions:** Conceptualization, Haozhe Cheng and Wanfeng Dou; Data curation, Haozhe Cheng; Formal analysis, Haozhe Cheng and Wanfeng Dou; Funding acquisition, Wanfeng Dou; Methodology, Haozhe Cheng; Software, Haozhe Cheng; Writing—original draft, Haozhe Cheng; Writing—review & editing, Haozhe Cheng and Wanfeng Dou. All authors have read and agreed to the published version of the manuscript.

**Funding:** This work was supported by the National Natural Science Foundation of China (No. 41930102, No. 41771411).

**Data Availability Statement:** All data are available online. Respective origins of these data are stated in this article.

**Acknowledgments:** The authors thank the reviewers for their patience and suggestions. They helped improve the article. The authors also thank Yuan Pang for suggestions on English grammar.

**Conflicts of Interest:** No potential conflict of interest was reported by the authors.

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
