# Peer review of "HiPDERL: An Improved Implementation of the PDERL Viewshed Algorithm and Accuracy Analysis"

_ijgi, doi:10.3390/ijgi11100506_

Round 1

Reviewer 1 Report

This study focused on HiPderl: An improved implementation of the PDERL viewshed 2 algorithm and its accuracy analysis. The article has scope to publish in the journal but not in the present form. The introduction needs refinements. Methodology is poorly written. Many sentences are unclear in the manuscript. The current version of the manuscript is written like a report summary. The conclusion should be specific. It is recommended to just highlight the key findings of the work. In the section Research gaps and future scope it should be specific not in general form. Figures and Tables need to be re-formulated. Results need lot of improvements. I would recommend for major corrections.

Comment 1: The abstract doesn’t show the accurate content and the main findings of the study area. Please add the main findings of the research work.

Comments 2: I recommend the authors to write in the Introduction more explicitly based on existing literature what is missing in previous studies, what is the added value of this new study.

Comments 3: The current version of the manuscript is written like a report summary. Please check and update it.

Comment 4: Please proofread the article carefully; there are many linguistic errors in the manuscript.

Comment 5: The methodology section is very unclear and needs updates. 

Comment 6: Update the figure 2b. The legibility is not good.

Comments 7: Please add more discussion material to the “Section 4 “Experimental Results” section. What were perhaps different results from other studies and why?

Comments 8: In the manuscript, Authors written the general conclusion.  The conclusion should be specific. It is recommended to just highlight the key findings of the work.

The article has the scope to publish in the journal but not in the present form. The introduction needs refinements. The methodology is poorly written. Many sentences are unclear in the manuscript. Figures and Tables need to be re-formulated. Results need a lot of improvements. I would recommend major corrections.

Author Response

Reviewer 1

This study focused on HiPderl: An improved implementation of the PDERL viewshed 2 algorithm and its accuracy analysis. The article has scope to publish in the journal but not in the present form. The introduction needs refinements. Methodology is poorly written. Many sentences are unclear in the manuscript. The current version of the manuscript is written like a report summary. The conclusion should be specific. It is recommended to just highlight the key findings of the work. In the section Research gaps and future scope it should be specific not in general form. Figures and Tables need to be re-formulated. Results need lot of improvements. I would recommend for major corrections.

Comment 1: The abstract doesn’t show the accurate content and the main findings of the study area. Please add the main findings of the research work.

Answer: Rework of the abstract is done. Thank you.

Comments 2: I recommend the authors to write in the Introduction more explicitly based on existing literature what is missing in previous studies, what is the added value of this new study.

Answer: Done. The original study of PDERL did not mention the cases that the observer is placed on grid points. In practice, PDERL did not work well on the cases.

Comments 3: The current version of the manuscript is written like a report summary. Please check and update it.

Answer: It might be better now, after revising.

Comment 4: Please proofread the article carefully; there are many linguistic errors in the manuscript.

Answer: Some errors are corrected.

Comment 5: The methodology section is very unclear and needs updates. 

Answer: It might be better now.

Comment 6: Update the figure 2b. The legibility is not good.

Answer: Done. Thank you.

Comments 7: Please add more discussion material to the “Section 4 “Experimental Results” section. What were perhaps different results from other studies and why?

Answer: Done.

Comments 8: In the manuscript, Authors written the general conclusion.  The conclusion should be specific. It is recommended to just highlight the key findings of the work.

Answer: Conclusion is rewritten.

The article has the scope to publish in the journal but not in the present form. The introduction needs refinements. The methodology is poorly written. Many sentences are unclear in the manuscript. Figures and Tables need to be re-formulated. Results need a lot of improvements. I would recommend major corrections.

Answer: All figures and tables are re-formulated. Some experiments are improved with a useful implement. The introduction is refined.

Reviewer 2 Report

Viewshed analysis is a basic task for digital terrain analysis. This paper presents a new algorithm HiPderl, which is basically an improved implementation algorithm of viewshed computation based on PDERL algorithm by the analysis of implementation flaws and float number accuracy problems affecting the computational accuracy. The experimental results verify the accuracy of the proposed algorithm compared with other viewshed algorithms. Some suggests are given as follows:

 1. The authors may add the full-name of all the abbreviation, for example, ‘proximity-direction-elevation’ (PDE) spatial reference line’ for PDERL. This will be helpful to the readers who are not familiar with the specific terms.

2. Moreover, the authors may use the original citations for the baseline algorithm PDERL. The following is for your reference.

Wu, C., Guan, L., Xia, Q., Chen, G., & Shen, B. (2021). PDERL: an accurate and fast algorithm with a novel perspective on solving the old viewshed analysis problem. Earth Science Informatics, 14(2), 619-632.

3. In section 3.2, the goal and relation of the Checking algorithm for PDERL with improved HiPderl algorithm needs to describe further. Besides, since the proposed algorithm is actually an enhanced PDERL, HiPDERL (with all capital letter) will be better to keep the unique by the original name.

4. Some formula symbols need to check carefully to avoid errors.

5. Some statements need to check so as to express clearly the ideas of the improved algorithm. For example, you may add a short paragraph from the very beginning of section 3 that why and how PDERL was enhanced by your proposed algorithm.

6. Moreover, the authors may polish the language to make the manuscript easy to follow. Typos should be also avoided.

7. As for the layout of the manuscript, figures and tables should be in one page.

Author Response

Reviewer 2

Viewshed analysis is a basic task for digital terrain analysis. This paper presents a new algorithm HiPderl, which is basically an improved implementation algorithm of viewshed computation based on PDERL algorithm by the analysis of implementation flaws and float number accuracy problems affecting the computational accuracy. The experimental results verify the accuracy of the proposed algorithm compared with other viewshed algorithms. Some suggests are given as follows:

  1. The authors may add the full-name of all the abbreviation, for example, ‘proximity-direction-elevation’ (PDE) spatial reference line’ for PDERL. This will be helpful to the readers who are not familiar with the specific terms.

Answer: Done. It is revised. Thank you.

  1. Moreover, the authors may use the original citations for the baseline algorithm PDERL. The following is for your reference.

Wu, C., Guan, L., Xia, Q., Chen, G., & Shen, B. (2021). PDERL: an accurate and fast algorithm with a novel perspective on solving the old viewshed analysis problem. Earth Science Informatics, 14(2), 619-632.

Answer: This is critical and revised. Thank you.

  1. In section 3.2, the goal and relation of the Checking algorithm for PDERL with improved HiPderl algorithm needs to describe further. Besides, since the proposed algorithm is actually an enhanced PDERL, HiPDERL (with all capital letter) will be better to keep the unique by the original name.

Answer: This is stated in the beginning of section 3, along with the statements mentioned in suggestion 5. HiPDERL is better. Thank you.

  1. Some formula symbols need to check carefully to avoid errors.

Answer: Checked.

  1. Some statements need to check so as to express clearly the ideas of the improved algorithm. For example, you may add a short paragraph from the very beginning of section 3 that why and how PDERL was enhanced by your proposed algorithm.

Answer: Done. Thank you.

  1. Moreover, the authors may polish the language to make the manuscript easy to follow. Typos should be also avoided.

Answer: Done.

  1. As for the layout of the manuscript, figures and tables should be in one page.

Answer: Checked.

Reviewer 3 Report

This paper sets out adjustments to the algorithm and implementation of PDERL which attempt to correct some sources of error, in particular accumulation of floating point errors, errors due to rounding close to zero and omitted calculation in certain circumstances.

The introduction and section 2.1 sets out the paper and reviews viewshed algorithms briefly. It is appropriate to be brief given the audience but I would suggest the division between between multiplex and none multiplex methods is secondary to that between raster based line of sight and object based graphic pipeline methods, but the literature selected doesn’t really mention work in the latter. This is relevant because the PDE method is in perspective projection so sits between the two.

Section 2 sets out how PDERL works and where the errors arise. In doing so it uses three figures from the original paper. Figure 1 and 2 are stated as coming from that paper but permissions are not given. Figure 5 is not acknowledged as being figure 7 from the original paper. I have not checked for text commonality but this needs to be done and the correct permissions stated.  I was left wondering how it handles edge cases due to cell granularity? That paper is not the one under review here but I think many readers might also like to know.

Ensure acronyms are defined before use (PDE).

Section 3 sets out the new algorithm to correct stated errors in the PDERL.

Error 1 > Random effects of values near zero.  This is a common, perhaps insurmountable, problem with perspective space methods for hidden surface removal. The suggested solution –line 324- appears to be simply to set the change in height to zero, thus the line can have no effect and is skipped. This removes the random result, but hides rather than corrects the error, because the true error is unknown.

Error 2>  Cumulative rounding due to floating points. The solution proposed here is to recalculate some aspects which were previously based on prior calculation, thus breaking transmission of error.  This is a valid approach, but as the section correctly concludes it trades efficiency for accuracy. Whether it is useful depends on the balance between them (see below).

Error 3 > Calculating viewpoints on the grid (Fig 6). I disagree with the statement on line 399 that  this does not belong to the original algorithm. The issue of being ‘neither left nor right’, is a consequence of using left and right in the logic of the algorithm.  So this is a substantial improvement to the PDERL algorithm for that reason and because the errors concerned are likely to be both spatially autocorrelated and periodic with the grid. However I am not quite convinced that this is solved without reliance on the solutions to the first two? Perhaps the authors could clarify this.

Section 4 sets out some results showing both where the errors are resolved (fig 8) and comparisons for accuracy (fig 9, table 1) and speed (table 2).  A few specific points:

Figure 8: Add the view point to all figures

Figure 9: Visual comparison is very hard. Try overlaying using colours to show the differences?

Tables 1 and 2 need the base line adding (i.e. how many cells in each  landscape type?).

From the data presented it is hard to assess whether the trade off between accuracy and speed discussed in section 3 are net beneficial in terms of “cells correctly categorised per second” for different landscape types.  Also needs discussion of where would one algorithm be faster or more accurate e.g. in what kinds of landscape? This is very hard to prove of course as shown by Peter Fisher and other’s early work on viewshed uncertainty (which isn’t mentioned in this paper). In particular, the original paper by Wu et al. (Ref 13) talks allot about continuous error, i.e. the spatial autocorrelation of the error but that is not calculated at all here.

That the authors study a previous work in detail and improve it is to be commended. Too often the goal is to create a “new” algorithm which simply adds another set of suboptimal solutions to what is a hard problem. However, my impression is that this paper sits uneasily between two ambitions. One is simply a correction to implementation errors within a published work, which is arguably better achieved as a letter to the original journal. The other is extending the original algorithm and showing it is more efficient and more accurate than other options, to which end it must first better define the application context . While it is faster than R3, for purposes where accuracy is paramount R3 remains substantially better. Where speed or extent is paramount it appears to have considerable advantages, the problem is the comparisons presented are for single viewpoints, not for total viewshed, intervisibilty or similarly demanding cases. Between these two is an optimal front where a certain level of accuracy and speed are both required. The question being what that threshold is and whether it is defined simply by global accuracy at the level of individual cells or if local clustering of error is important.

Author Response

Reviewer 3

This paper sets out adjustments to the algorithm and implementation of PDERL which attempt to correct some sources of error, in particular accumulation of floating point errors, errors due to rounding close to zero and omitted calculation in certain circumstances.

The introduction and section 2.1 sets out the paper and reviews viewshed algorithms briefly. It is appropriate to be brief given the audience but I would suggest the division between between multiplex and none multiplex methods is secondary to that between raster based line of sight and object based graphic pipeline methods, but the literature selected doesn’t really mention work in the latter. This is relevant because the PDE method is in perspective projection so sits between the two.

Answer:

The original division is actually between R3 and others, in order to show that the PDERL algorithm is somehow like R3, but more efficient as others be. They are now renamed as the accurate one and the efficient ones. Maybe this could eliminate some confusion.

The work of Tabik et al. (2013) along with the following work is now mentioned in the introduction as well.

Tabik et al. (2013) https://dx.doi.org/10.1080/13658816.2012.677538

Section 2 sets out how PDERL works and where the errors arise. In doing so it uses three figures from the original paper. Figure 1 and 2 are stated as coming from that paper but permissions are not given. Figure 5 is not acknowledged as being figure 7 from the original paper. I have not checked for text commonality but this needs to be done and the correct permissions stated.  I was left wondering how it handles edge cases due to cell granularity? That paper is not the one under review here but I think many readers might also like to know.

Answer:

Permission from the original authors are now stated.

PDERL does the same work as R3 does, but in a reorganized way. On this point they might be the same.

Ensure acronyms are defined before use (PDE).

Answer:Done. Thank you.

Section 3 sets out the new algorithm to correct stated errors in the PDERL.

Error 1 > Random effects of values near zero.  This is a common, perhaps insurmountable, problem with perspective space methods for hidden surface removal. The suggested solution –line 324- appears to be simply to set the change in height to zero, thus the line can have no effect and is skipped. This removes the random result, but hides rather than corrects the error, because the true error is unknown.

Answer:

As an example of the errors, (2.5×0.4 - 10×0.1) is not calculated as absolute zero but a small positive value. This might be set to zero. The criteria for a small value to be set to zero is 10^-12. This may correct very small values made by machine operation only.

Error 2>  Cumulative rounding due to floating points. The solution proposed here is to recalculate some aspects which were previously based on prior calculation, thus breaking transmission of error.  This is a valid approach, but as the section correctly concludes it trades efficiency for accuracy. Whether it is useful depends on the balance between them (see below).

Answer:

Solutions to float numbers problems seem separate, but they all involve small float absolute values, and may interfere each other. It might be better to regard them as one in experiments.

The original PDERL for experiments will be without correction on error avoiding mechanism now. The remade error avoiding mechanism is part of float number problem solutions after all. New results are more considerate and of course real.

Error 3 > Calculating viewpoints on the grid (Fig 6). I disagree with the statement on line 399 that  this does not belong to the original algorithm. The issue of being ‘neither left nor right’, is a consequence of using left and right in the logic of the algorithm.  So this is a substantial improvement to the PDERL algorithm for that reason and because the errors concerned are likely to be both spatially autocorrelated and periodic with the grid. However I am not quite convinced that this is solved without reliance on the solutions to the first two? Perhaps the authors could clarify this.

Answer:

Solution to this problem is completely irrelevant with others, since these target points never get the chance of being visible by the original implementation. In another word, they are out of the PDERL algorithm. For improvement, it would be like processing the target points within the PDERL system, and then processing the unprocessed column independently.

Section 4 sets out some results showing both where the errors are resolved (fig 8) and comparisons for accuracy (fig 9, table 1) and speed (table 2).  A few specific points:

Figure 8: Add the view point to all figures

Answer: Done. Part of the figures are amplified.

Figure 9: Visual comparison is very hard. Try overlaying using colours to show the differences?

Answer:Part of the figures are amplified. Should make the differences easier to see.

Tables 1 and 2 need the base line adding (i.e. how many cells in each  landscape type?).

Answer:Done. Thank you.

Reviewer 4 Report

Manuscript "HiPderl: An improved implementation of the PDERL viewshed algorithm and accuracy analysis" presents an implementation of a line-of-sight (LoS) algorithm that is based on the coordinate transformation of grid points onto a proximity-direction-elevation descriptor. Overall the description is correct and informative.

Abstract:

* Present briefly PDERL before mentioning it in the abstract, such as "PDERL is an algorithm that presents computational advantages when compared to other multiplexing algorithms, while keeping accuracy" 

Introduction:

* 1st paragraph is confusing, why introducing a difference between "viewshed" and "terrain viewshed analysis"?

* Besides the obvious direct viewshed analytical applications, the binary result of the calculation can be combined with other grids to describe terrain. See e.g. https://doi.org/10.1007/s10761-016-0334-9 or https://doi.org/10.1016/j.jas.2015.06.015

Experimental results: 

* Please provide details such as spatial resolution of the datasets used to test the algorithm

* Please describe the morphology of the area around viewpoint located at row 96, column 176 on the DEM of Malaga: local maximum surrounded by higher hills, valleys, ridges, etc.

* Table 1: please add fractions (%) in order to estimate the % of error, along with the number of cells already mentioned

* Table 1: I don't know if it is meaningful to insert the XDraw and HiXDraw values here, as they belong to a different category than the PDERL and HiPderl formulations. The paper presents an implementation of an improvement of the PDERL algorithm, which itself was an order of magnitude better than the XDraw family of formulae, and it is with this PDERL algorithm that is should be compared to.

* The images of Fig. 9 do now allow to understand if there is a spatial pattern for the most frequent differences between PDERL and HiPderl. Malaga is a coastal city, and the most visible differences are in the sea part. However, as the authors have classified the terrain as hills, mountains and plains, it would be interesting to understand where are the main differences (plains? plains behind close hills? something in terms of proximity/elevation?). A table or map with the differences would be very informative, so one can decide, e.g., to use this implementation instead of PDERL if it performs better in very mountainous terrain.

Minor English language issues:

Abstract: 

"respectively focus on"-->"respectively focusing on"

"taking the both advantages" --> "taking advantages from both"

Introduction:

"doing so, The viewshed" --> "doing so, the viewshed"

"basis of the most" --> "basis of most"

"respectively focus on"-->"respectively focusing on"

"as an example, the XDraw algorithm takes" --> "as an example, it takes"

"results by a data" --> "results in a data"

"thus solves the problem" --> "thus solving the problem"

Related works:

"Xdraw algorithm records" --> "XDraw algorithm records"

Author Response

Reviewer 4

Manuscript "HiPderl: An improved implementation of the PDERL viewshed algorithm and accuracy analysis" presents an implementation of a line-of-sight (LoS) algorithm that is based on the coordinate transformation of grid points onto a proximity-direction-elevation descriptor. Overall the description is correct and informative.

Abstract:

* Present briefly PDERL before mentioning it in the abstract, such as "PDERL is an algorithm that presents computational advantages when compared to other multiplexing algorithms, while keeping accuracy" 

Answer: Done. It is revised. Thank you.

Introduction:

* 1st paragraph is confusing, why introducing a difference between "viewshed" and "terrain viewshed analysis"?

Answer: That is quite confusing. Expressions are changed.

* Besides the obvious direct viewshed analytical applications, the binary result of the calculation can be combined with other grids to describe terrain. See e.g. https://doi.org/10.1007/s10761-016-0334-9 or https://doi.org/10.1016/j.jas.2015.06.015

Answer:Added into introduction. Thank you.

Experimental results: 

* Please provide details such as spatial resolution of the datasets used to test the algorithm

Answer:Done. Thank you.

* Please describe the morphology of the area around viewpoint located at row 96, column 176 on the DEM of Malaga: local maximum surrounded by higher hills, valleys, ridges, etc.

Answer:Done. It is implemented by some experiments.

* Table 1: please add fractions (%) in order to estimate the % of error, along with the number of cells already mentioned

Answer:Done. It is revised.

* Table 1: I don't know if it is meaningful to insert the XDraw and HiXDraw values here, as they belong to a different category than the PDERL and HiPderl formulations. The paper presents an implementation of an improvement of the PDERL algorithm, which itself was an order of magnitude better than the XDraw family of formulae, and it is with this PDERL algorithm that is should be compared to.

Answer:

XDraw and HiXDraw are used as references, in order to show the results of a traditional multiplexing algorithm and its enhanced version. This statement is added into the paper.

The original research used XDraw and Reference Plane algorithm for experiment as well.

* The images of Fig. 9 do now allow to understand if there is a spatial pattern for the most frequent differences between PDERL and HiPderl. Malaga is a coastal city, and the most visible differences are in the sea part. However, as the authors have classified the terrain as hills, mountains and plains, it would be interesting to understand where are the main differences (plains? plains behind close hills? something in terms of proximity/elevation?). A table or map with the differences would be very informative, so one can decide, e.g., to use this implementation instead of PDERL if it performs better in very mountainous terrain.

Answer:

The DEM of Malaga is only for better demonstration of the problems. On this DEM the consequence of this problem would be more clear to see. This statement is added into the paper.

The original PDERL is now without solutions to some problems, because these problems are seen as part of PDERL origin idea, and also may interfere changes within HiPDERL.

Due to the design of the experiment, the viewpoints can be further divided according to error rate. Data added into the paper.

Minor English language issues:

Abstract: 

"respectively focus on"-->"respectively focusing on"

"taking the both advantages" --> "taking advantages from both"

Introduction:

"doing so, The viewshed" --> "doing so, the viewshed"

"basis of the most" --> "basis of most"

"respectively focus on"-->"respectively focusing on"

"as an example, the XDraw algorithm takes" --> "as an example, it takes"

"results by a data" --> "results in a data"

"thus solves the problem" --> "thus solving the problem"

Related works:

"Xdraw algorithm records" --> "XDraw algorithm records"

Answer:All corrected. Thank you.

Round 2

Reviewer 1 Report

Authors have made all the required changes. Hence, the revised manuscript can be accepted.

Author Response

Reviewer 1

Authors have made all the required changes. Hence, the revised manuscript can be accepted.

Answer:

Thank you for your patience and suggestions.

Reviewer 3 Report

Line 175, author permission is not sufficient you need a licence # from Springer. Also, its not sufficient to state permission once, it should be in every figure caption.

"completely irrelevant as .. they are out of the PDERL algorithm. 403" 

"To solve this  problem, independent visibility computation of the target points mentioned above is added to the normal workflow of PDERL in this paper" 400

It may be a "null" cell in PDERL but that isnt irrelevant when comparing with R3 which does "see" those points. Please state more clearly what the independent method used to compute these lines of site was, e.g. R3? 

4.1 The text is OK, the figures need some minor adjustments. I can see this is not easy because R3 and HiPDERL give similar results so you cannot show both atop one another when there are no visible differences. But you can still state which is the source for the image in the figure then simply state that the other is not visibly different (I assume individual cell differences are scattered so not visible at that scale?).

The figures also need a legend, scale and north arrow. 

A context map showing the shaded relief would also help, or a 3D scene from the viewpoint.

4.2.3. Table 1 is much improved. It makes a good case for the accuracy of HiPDERL although it would be more convincing still if there were some statistics on e.g. patch size or autocorrelation of these error cells (one big error is much more problematic than many individual cells randomly scattered).

Over all, with these minor corrections I can recommend acceptance (editorial staff can decide if the corrections are done, I do not need to see the new version). The English is OK but rather informal in places so a language check would be wise.

Author Response

Reviewer 3

Line 175, author permission is not sufficient you need a licence # from Springer. Also, its not sufficient to state permission once, it should be in every figure caption.

Answer:

We redraw all pictures from the original paper by our ideas.

"completely irrelevant as .. they are out of the PDERL algorithm. 403" 

"To solve this  problem, independent visibility computation of the target points mentioned above is added to the normal workflow of PDERL in this paper" 400

It may be a "null" cell in PDERL but that isnt irrelevant when comparing with R3 which does "see" those points. Please state more clearly what the independent method used to compute these lines of site was, e.g. R3? 

Answer:

This is critical. PDERL initializes all target points as invisible. Since the line of target points are never processed, they will remain invisible. For HiPDERL, the independent method is R3, but data reuse is available because the target points are in a line. This is mentioned in the article. Thank you.

4.1 The text is OK, the figures need some minor adjustments. I can see this is not easy because R3 and HiPDERL give similar results so you cannot show both atop one another when there are no visible differences. But you can still state which is the source for the image in the figure then simply state that the other is not visibly different (I assume individual cell differences are scattered so not visible at that scale?).

The figures also need a legend, scale and north arrow. 

A context map showing the shaded relief would also help, or a 3D scene from the viewpoint.

Answer:

The result by R3 and HiPDERL are the same by every point. The 200x200 result maps are checked. Maybe this should be mentioned. Others are done.

4.2.3. Table 1 is much improved. It makes a good case for the accuracy of HiPDERL although it would be more convincing still if there were some statistics on e.g. patch size or autocorrelation of these error cells (one big error is much more problematic than many individual cells randomly scattered).

Answer:

Discussion on the gathering or scattering error points is added.

Over all, with these minor corrections I can recommend acceptance (editorial staff can decide if the corrections are done, I do not need to see the new version). The English is OK but rather informal in places so a language check would be wise.

Answer:

The paper is checked carefully.
